# "I just don't know what will be better other than an apartment which I can't afford": Experiences of housing quality following homelessness in Ontario, Canada

Carrie Anne Marshall◉*, Patti Plett◉, Jessica Allen◉,
Corinna Easton, Rebecca Goldszmidt◉, Elham Javadizadeh◉, Shauna Perez◉,
Brooklyn Ward

Social Justice in Mental Health Research Lab, School of Occupational Therapy, Western University, London, Ontario, Canada

* carrie.marshall@uwo.ca

## Abstract

Having access to good quality housing is a key determinant of well-being. Little is known about experiences of housing quality following homelessness from the perspectives of persons with lived experience. To build on existing literature, we conducted a secondary analysis of qualitative interviews with 19 individuals who had experiences of transitioning to housing following homelessness. Interview transcripts were drawn from a community-based participatory research study exploring the conditions needed for thriving following homelessness in Ontario, Canada. We analyzed these transcripts using reflexive thematic analysis. We coded transcripts abductively, informed by theories of social justice and health equity. Consistent with reflexive thematic analysis, we identified a central essence to elucidate experiences of housing quality following homelessness: "negotiating control within oppressive structural contexts." This was expressed through four distinct themes: 1) being forced to live in undesirable living conditions; 2) stuck in an unsafe environment; 3) negotiating power dynamics to attain comfort and safety in one's housing; and 4) having access to people and resources that create home. Overall, our findings indicate that attaining good quality housing following homelessness is elusive for many and influenced by a range of structural factors including ongoing poverty following homelessness, a lack of deeply affordable housing stock, and a lack of available social support networks. To prevent homelessness, it is essential to improve access to good quality housing that can support tenancy sustainment and well-being following homelessness. Policymakers need to review existing housing policies and reflect on how over-reliance on market housing has imposed negative impacts on the lives of persons who are leaving homelessness. Given the current economic context, it is imperative that policymakers devise policies that mitigate the financialization of housing, and result in the restoration of the social housing system in Canada and beyond.

**Data availability statement:** Data will be made available upon reasonable request. Data cannot be added to a repository as data is potentially identifying, and would violate confidentiality of participants. Further, providing the data openly would violate the ethics protocols that have been approved by the research ethics boards at Western and Queen's Universities. Data requests can be posed by contacting the Western University Research Ethics Board at: wrem@uwo.ca.

**Funding:** This study was funded by the Canadian Institutes of Health Research in the form of a Project Grant awarded in to the principal investigator in 2019 [Grant number: PJT 166132]. The funder had no role in study design, data collection and analysis, decision to publish, or preparation of the manuscript.

**Competing interests:** The authors have declared that no competing interests exist.

## Introduction

Healthy and good quality housing is an important determinant of human health and well-being [1] and is something that individuals often struggle to obtain during and following homelessness [2–4]. Poor quality housing has been associated with a range of negative health consequences including lowered mental health across the lifespan [5], poor respiratory health [6], and increased risk of physical injury [7]. Rising real estate values in recent decades [8] combined with dwindling government investment and chronic underfunding of social housing [9] has created a severe shortage of deeply affordable housing available for individuals living in low income. The dire lack of market and social housing units resulting from these conditions enables unscrupulous private landlords to rent sub-standard units to prospective tenants and limits the ability of social housing providers to properly maintain aging housing stock. As a result, individuals who experience poverty, including persons who experience homelessness, are especially vulnerable to living in poor quality housing. Often, the only housing that is available to individuals living on the lowest incomes in society is that which places their health and well-being at risk [4]. As such, individuals living in poverty often face an impossible situation of being forced into living in poor quality housing as they leave homelessness, and tolerate these conditions to remain housed. Exclusion from access to good quality housing is a serious human rights violation that not only deepens the inequities that persons living in poverty already experience, but also perpetuates housing precarity and homelessness [10,11].

### Housing quality in relation to homelessness prevention

While significant efforts have been dedicated to addressing homelessness internationally, it is a phenomenon that continues to grow. It has been estimated that globally, 150 million individuals experience homelessness every year, with 15 million forcefully evicted from their homes and a further 1.6 billion people living in inadequate housing conditions [12]. It is generally agreed upon by scholars and practitioners that a key strategy for preventing and ending homelessness is to provide permanent housing coupled with individualized support for tenancy sustainment and attaining the necessary conditions for social, physical, emotional, and economic well-being [13]. Research exploring the effectiveness of an intervention that translates this approach into practice, Housing First, demonstrates that it is more effective than standard models of care at supporting tenancy sustainment and reducing the utilization of costly forms of health care such as in-patient stays and emergency services [14]. As such, Housing First is regarded as a necessary homelessness prevention approach.

Tenancy sustainment is a critical outcome of any strategy focused on homelessness prevention. It is intimately related to housing quality, with existing research identifying housing quality as a predictor of whether a person is able to sustain a tenancy [15]. Further, while the quality of one's housing is frequently identified as an important determinant of well-being following homelessness, many individuals identify that they struggle to attain housing that provides adequate living conditions following homelessness [3,4,16]. Research on the transition to housing indicates that even when individuals are able to secure and sustain a tenancy, that many indices

of psychosocial well-being are poorly addressed, leaving some languishing in their housing after leaving homelessness [3,17,18]. Outcomes that are poorly addressed following homelessness include substance use [14], engagement in meaningful activity and employment [19–21], and community and social integration [18,22,23]. A lack of housing quality may contribute to or complicate the inability to attain these important outcomes.

### Research focused on housing quality following homelessness

A recent scoping review synthesizing literature on housing quality and its relationship with psychosocial well-being following homelessness identified 32 articles representing various research methods [4]. This review provides evidence of the relationship between housing quality and well-being among previously unhoused persons and identifies that housing options are often limited to lower quality housing as individuals transition from living in shelters or the street. Few of the studies included in this review, however, focused on housing quality as a central construct of interest, with housing quality emerging only as a secondary finding [4]. Most of the included studies in this review were quantitative, and as such, explored correlations between housing quality and indices of psychosocial well-being. Only one of the studies included in this review *focused* on the experience of housing quality using qualitative methods [24]. The physical state of participants' housing, as well as the location of their housing were identified as key factors that mitigated participants' experiences [24]. This study was conducted in the United States in a permanent supportive housing program and focused specifically on how the state of one's housing influenced quality of life. There are no studies known to our team that have focused on the experience of housing quality following homelessness in relation to psychosocial well-being across a range of housing contexts including market units, transitional housing, permanent supportive housing, social housing, and others. Further, there is a need to build on existing research conducted in the United States to further understand the experience of housing quality for persons leaving homelessness in other countries. Filling these gaps in existing research literature is critical for informing future research, policy and practice to support well-being and tenancy sustainment of individuals who are housed following homelessness.

### The current study and theoretical framework

We conducted this study to identify the experience of housing quality and its relationship with psychosocial well-being following homelessness in a Canadian context with participants living in a range of housing situations. This research was guided by theoretical lenses of social justice and health equity. Social justice theories advocate for equitable access to resources for all individuals irrespective of their class, race, gender, sexual orientation or other social locations [25]. Health equity is a form of social justice that seeks to advocate for what specific individuals need to thrive in their lives given their available resources and abilities [25]. Identifying experiences of housing quality following homelessness has the potential to further an understanding of the lived realities of individuals who are leaving homelessness, and use this evidence to advance targeted policy and practice developments aimed at improving the living conditions of this population. The research question guiding this study was: "What is the experience of housing quality in relation to psychosocial well-being following homelessness?"

## Materials and methods

### Ethic Statement

Ethics approval was obtained by the authors of the parent study from the Western University Non-Medical Research Ethics Board (#114992) and Queen's University Health Sciences Research Ethics Board (#6028323) in London and Kingston, Ontario, respectively. A separate ethics application was not required for the current study, as participants in the parent study provided approval for the conduct of secondary analyses of pseudonomized transcripts. We conducted a secondary analysis of qualitative interviews derived from a stakeholder consultation conducted in the context of a community-based participatory research (CBPR) [26] study aimed at identifying what conditions are needed to enable thriving following

homelessness. This study, called the "Transition from Homelessness Study" [2,3,27,28], was conducted in two cities, and was designed to inform the development of approaches for supporting 'thriving' rather than solely outcomes associated with 'survival' following homelessness. Participants in the parent study discussed their experiences of housing quality after leaving homelessness at length, and conducting a secondary analysis of this data presented an opportunity to share participants' experiences more fulsomely on this topic.

## Recruitment

In the parent study, participants were recruited from organisations providing social service and mental health support to persons with lived experiences of homelessness in Kingston and London, Ontario, Canada. We used a purposive sampling strategy in the parent study to generate a sample of individuals who had recent experiences of homelessness. To facilitate recruitment, we arranged suitable times to come to shelters, drop-in centres, encampments, and housing programs with staff in the recruitment organizations. During these times, a research assistant occupied a private interview space where potential participants could ask questions about the study, and where a research assistant could determine eligibility. Further, staff in shelters, drop-in centres, and housing programs were invited to provide the contact of the research team to persons with experiences of homelessness and they were invited to reach out to the team directly via email, telephone or text. None of the participants meeting inclusion criteria were excluded from the study following recruitment.

Participants were included if they were over the age of 16, and had experienced at least one month of homelessness in the past three years, irrespective of whether they were currently housed or unhoused during interviews. All participants were provided with $40 in cash as an honorarium for their involvement in interviews. Only a sub-sample of participants who had been housed following homelessness from the initial sample was included in the secondary analysis presented in the current paper. Following recruitment, all participants were read a letter of information aloud by a member of the research team, were provided an opportunity to ask questions about the study, and then signed a form indicating their consent to participate. Each participant was asked to provide a pseudonym to protect their confidentiality, which have been used to label quotes in the results section, rather than their real names. A list linking the participant's pseudonym with their real name was kept by the principal investigator of the parent and the current study, which was kept confidential throughout the research process, by limiting access only to the research team involved in the parent study. A more detailed description of the processes used to recruit participants in the parent study is provided in a separate publication [3].

## Data collection

Participants who met the inclusion criteria of the parent study arranged suitable times with research staff to participate in interviews. Interviews were semi-structured and consisted of questions focused on the conditions needed for individuals to "thrive" following homelessness. Researchers conducting the parent study also collected demographic information, including details about the mental health of participants. This included the delivery of two standardized measures of substance use including the Alcohol Use Disorder Identification Test-10 (AUDIT-10) [29] and Drug Abuse Screening Test-10 (DAST-10) (30). The AUDIT-10 is a 10-item standardized measure of alcohol use using a 3–5 point nominal scale. The DAST-10 is a 10-item standardized measure of substance use, excluding alcohol, tobacco and caffeine, on a dichotomous (yes/no) scale. Both of these measures have been broadly utilized in health and social care research, and have been demonstrated to be reliable scales with the AUDIT-10 demonstrating internal consistency of 0.75-0.97 in previous research [30] and the DAST-10 demonstrating internal consistency of 0.86-0.90 in previous research [31,19]. Interviewers were research assistants who had received training in the conduct of semi-structured qualitative interviews by the principal investigator and co-investigators of the parent study. Research assistants were persons with lived experiences of homelessness, individuals with experiences of providing service in social service and/or mental health roles with unhoused

persons, and/or research students at the master's or PhD-level who were enrolled in educational programs pertaining to health and rehabilitation fields. Interviews were recorded on a digital recording device and transcribed verbatim. Sample interview questions posed to participants are provided in Table 1.

## Analysis

We took an interpretivist approach to our analysis and coded interview transcripts abductively, informed by theories of social justice and health equity [25] using reflexive thematic analysis [32]. Abductive coding is a type of coding that explicitly acknowledges a researcher's theoretical positioning during the process of assigning codes to qualitative data [33]. This approach contrasts with inductive coding, which typically involves entering into a dataset with no prior theoretical understandings of the data, or deductive coding, where a theory is used to structure analysis of qualitative data [33]. Interview transcripts were uploaded to Dedoose, a cloud-based qualitative data management program that facilitated the organisation of our data [34]. Members of our research team (CM, CE, BW, PP, JA, SP, RG, EJ) coded statements from the interview transcripts that pertained to the experience of housing quality in relation to psychosocial well-being following homelessness. Codes were then grouped into categories and categories were arranged into themes. Themes were developed and refined through group discussion and consensus. Consistent with the method advanced by Braun and Clarke, we identified a central essence of the experiences of participants expressed through the themes generated in our analysis [32]. Once our findings were analysed and written, all study authors provided final feedback on the written analysis, and our themes and essence were refined further.

In both the current and the parent study, we did not engage in member checking, as in the experience of the principal investigator, locating participants with experiences of homelessness on more than one occasion can be difficult due the ongoing changes that tend to occur in their lives. Participants who consented to be contacted with the results of the study were provided with a copy of reports and publications resulting from this research using the contact information provided during interviews. Further, organizations that provided services to persons experiencing homelessness in the recruitment cities were provided with all reports and published papers and encouraged to distribute among staff and service users. Further, public presentations were provided in-person and virtually throughout the communities in which we recruited that participants who consented to participate and staff and leadership in organizations that provided services to persons experiencing homelessness were invited to attend.

**Table 1. Sample semi-structured interview questions.**

1. Tell me about your experiences of getting your own place after living in shelters/on the street.
2. When you were making the transition back into your own place, was there anything that you needed individually that you received from family/friends, service providers, or others in the community that you found especially helpful? What were they?
3. Was there anything that you felt you needed, but didn't receive from family/friends, service providers, or others in the community? What were they?
4. Are there services that you wish you had access to that you didn't have when you moved into your place, or that you don't have now? What are they?
5. Are you receiving services that are particularly helpful/unhelpful? What specifically is helpful/unhelpful about these services?
6. What about your housing may support or detract from your mental well-being?
7. Are there times when you feel like you just want to go back to the shelter or live on the street? Why or why not?
8. Have you been able to find or keep family and friends that are good for your mental well-being? What has helped you to do this?
9. How do you spend much time doing activities in the community outside of your place, if at all?
10. In what ways do you feel like you belong in your community or not?
11. What do you need to be mentally well and feel like you're thriving when you're housed?
12. If you've quit using alcohol or drugs or have started to use more safely since becoming housed, what is helping you do that?
13. Is there anything we didn't ask about how you could be better supported to thrive in your housing?

## Trustworthiness

For the parent study, we employed strategies for establishing trustworthiness as identified by Lincoln and Guba [35], which were followed throughout the current study, including: 1) prolonged engagement with the population of interest, which was achieved through the research team's extensive involvement in research and practice related to homelessness, as well as lived experience of some team members with housing precarity and living in poor quality housing at various points in their lives; 2) peer debriefing, which involved continuous debriefing among all members of the research team during analysis; 3) recording interviews, which was completed in the primary collection of data; 4) accurate transcription, which had occurred prior to the conduct of this secondary analysis; 5) intercoder consensus; and 6) use of a computer program to organize data (Dedoose), which contributed to the dependability of our analysis.

## Reflexivity

Collectively, our team has several years of experience working with persons who have transitioned to housing following homelessness. It should be noted that the principal investigator of the current study was the study lead for the parent study, and that other members of our research team were involved in data collection and analysis (RG, CE, SP). Additionally, some of the members of our research team acknowledge having experiences with housing precarity or have lived in poor-quality housing in their lives. We take the position that these experiences have allowed us to better understand the experience of living in poor-quality housing following homelessness. While our experiences are not the same as the participants in this secondary analysis, we believe that our collective understandings have enabled us to notice important nuances of participant narratives that we would not have otherwise noticed without these personal and professional experiences. Thus, instead of bracketing, we have chosen to let our experiences shape our interpretations during the analytic process.

## Results

### Participant characteristics

Interviews from 19 participants were included in this secondary analysis, and ranged from 11-82 minutes in duration (Mdn = 42; IQR = 28). Participants included n = 9 men (47.4%), n = 9 women (47.4%), and n = 1 non-binary person (5.3%). Regarding sexual orientation, n = 3 participants identified as bisexual (15.8%), n = 2 as transsexual (10.5%) and n = 1 as queer (5.3%). The racial characteristics of participants were primarily White (n = 15; 78.9%), followed by First Nations (n = 2; 10.5%), Black (n = 1; 5.3%), and n = 1 preferred not to answer (5.3%). The most common source of income reported by participants was disability-related social assistance (Ontario Disability Assistance Program) (n = 14; 73.7%). A more fulsome description of the demographic characteristics of participants is provided in Table 2.

All participants included in this secondary analysis were housed following homelessness. The duration of their housing tenure at the time of interviews ranged from 1-36 months (Mdn = 12; IQR = 16). Over half were housed in market rental units (n = 10; 52.6%), followed by social housing (n = 5; 26.3%) and permanent supportive housing (n = 4; 21.1%). The age at which they first lost their housing ranged from 14-57 (Mdn = 23; IQR = 13.25).

Regarding mental health status, the most common conditions reported by participants included mood disorder (n = 94.7%), anxiety disorder (n = 94.7%), and stress and trauma-related disorders (n = 15; 78.9%). A total of seven (36.8%) participants reported alcohol use at a hazardous level according to the Alcohol Use Disorders Identification Test (AUDIT-10) (29). Over half of participants reported that the use of substances other than alcohol was non-problematic (n = 10; 52.6%), while n = 5 (26.3%) reported 'moderate' use, and n = 4 (21.1%) reported 'substantial' use on the Drug Abuse Screening Test (DAST-10) (30). See Table 3 for a detailed summary of the housing and health characteristics of participants included in this analysis.

**Table 2. Participant characteristics (n = 19).**

| Demographic Characteristics | |
| --- | --- |
| | **n (%)** |
| Gender | |
| Men | 9 (47.4) |
| Women | 9 (47.4) |
| Non-binary | 1 (5.3) |
| Age | (23–64; Mdn = 35; IQR = 19) |
| Sexual orientation | |
| Heterosexual | 13 (68.4) |
| Bi-sexual | 3 (15.8) |
| Transsexual | 2 (10.5) |
| Queer | 1 (5.3) |
| Race/Ethnicity | |
| White | 15 (78.9) |
| Black | 1 (5.3) |
| First Nations[1] | 2 (10.5) |
| Prefer not to answer | 1 (5.3) |
| Income source[2] | |
| ODSP | 14 (73.7) |
| Self-employment (e.g., bottle collecting, babysitting, shoveling snow, panhandling, etc.) | 6 (10.5) |
| Employment | 5 (26.3) |
| OW | 2 (10.5) |
| CPP | 1 (5.3) |
| OSAP | 1 (5.3) |
| Long term disability through former employer | 1 (5.3) |
| Public speaking/ lived experience consultant | 1 (5.3) |
| Work in supportive housing | 1 (5.3) |
| DSO | 1 (5.3) |

Note: ODSP = Ontario Disability Support Program; OW = Ontario Works; CPP = Canada Pension Plan; DSO = Developmental Services Ontario; OSAP = Ontario Student Assistance Program

[1](Cree = 1; Mohawk = 1)

[2]Frequencies exceed the total number of participants due to reports of more than one income source

## Thematic analysis

**Essence: Negotiating control within oppressive structural contexts.** The overarching essence expressed by participants included in this secondary analysis was that of negotiating control within structural contexts. Participants described how when they left homelessness, their choices in what housing was available to them was limited by their low incomes and an overall lack of good quality, affordable housing stock in the communities in which they lived. One participant described how he was offered two choices in housing as he was leaving homelessness including a poorly maintained unit or sharing a residence with a person living with substance use disorder. He chose the poorly maintained housing, which he described as "*toxic*": "*…I've been forced into another toxic situation. I chose it, but I was left to choose that as opposed to moving in with a drug addict*" [Pekoe]. Other participants expressed struggling with similar challenges, highlighting how they were forced to choose between equally problematic options. Although they were living in housing with bedbugs and roommates who did not clean up after themselves, one participant highlighted that it was better than

**Table 3. Participant housing history and health status (n = 19).**

| Characteristic | |
|---|---|
| | n (%) |
| How many months have you been housed? | (1–36; Mdn = 12; IQR = 16) |
| Months unhoused before securing a tenancy | (1–54; Mdn = 12; IQR = 22) |
| Housing type | |
| Market rent unit | 10 (52.6) |
| Social housing | 5 (26.3) |
| PSH (Cluster site) | 4 (21.1) |
| At what age did you first experience homelessness? | (14–57; Mdn = 23; IQR = 13.25) |
| Mental health conditions | |
| Mood disorder | 18 (94.7) |
| Anxiety disorder | 18 (94.7) |
| Stress and trauma disorder | 15 (78.9) |
| Obsessive-compulsive disorder | 8 (42.1) |
| Psychotic disorder | 6 (31.6) |
| Personality disorder | 6 (31.6) |
| Eating disorder | 1 (5.3) |
| Substance Use | |
| Alcohol use (AUDIT) | 0-40 (Mdn = 2; IQR = 10) |
| Non-hazardous use (<8) | 12 (63.2) |
| Hazardous use (≥8) | 7 (36.8) |
| Drug use (DAST) | 0-8 (Mdn = 0; IQR = 5) |
| No problem (0) | 10 (52.6) |
| Low level (1–2) | 0 (0) |
| Moderate level (3–5) | 5 (26.3) |
| Substantial level (6–8) | 4 (21.1) |
| Severe level (9–10) | 0 (0) |

**Note:** PSH = Permanent supportive housing; AUDIT = Alcohol Use Disorders Identification Test;
DAST = Drug Abuse Screening Test; Mdn = Median; IQR = Interquartile range.

their previous housing where "*safety was a real concern*" [Doc]. This same participant expressed his frustration, indicating: "*I just don't know what will be better other than an apartment which I can't afford*" [Doc]. Another participant who had chosen to live in a slightly more expensive building in an apartment that met their needs indicated that they made this decision by foregoing other necessary comforts such as furniture, which imposed a negative influence on their mental well-being: "*you don't have any furniture…that can put you in a really depressive state*" [Michelle]. While participants had limited choices in their housing, they were constantly negotiating control in how and where they lived within the context of structural limitations such as the presence of poverty and living in communities where the only affordable rental housing available was of poor quality. This essence was expressed through four themes that we generated in our analysis: 1) being forced to live in undesirable living conditions; 2) stuck in an unsafe environment; 3) navigating power dynamics to attain comfort and safety in one's housing; and 4) having access to people and resources that create home. See Fig 1 for a visual summary of our themes and relationships between the essence and the themes that comprise it.

**Theme 1: Being forced to live in undesirable living conditions.** Following homelessness, participants identified that the only housing options available to them were not ideal but that these options were all that they could afford. While most recognized their housing as poor quality, leaving them feeling uncomfortable and worried about their well-being,

the lack of housing available to them due to inadequate incomes provided few alternatives. Michelle described how moving to another housing unit was *"...just not possible so that's one of the main reasons why I stay where I am.*" Overall, participants identified feeling stuck in their housing, which they described as undesirable, and exposed them to conditions that had a profound influence on their psychosocial well-being. These conditions included the presence of pests, poorly maintained units and overcrowding.

Many participants described how they were forced to contend with bed bugs and rodents which imposed a range of risks to their well-being. One participant described an ongoing rat issue that went unresolved even after reporting it to the landlord soon after moving in to their apartment: *"the constant rats... they just keep coming back in the house*" [Perseus]. Left with few other options, Perseus dealt with this persistent rat infestation by storing his food in places such as the refrigerator to both protect his food from contamination, as well as deter the rats from remaining in his unit. Other participants described living with bedbugs on an ongoing basis, which one participant avoided by sleeping on his couch instead of his bed, where the infestation seemed to persist: *"there's bed bugs... I've been sleeping on the couch*" [Doc]. The presence of bedbugs prevented participants from feeling comfortable in their housing and attaining adequate sleep because they were awoken from being bit by these bugs throughout the night.

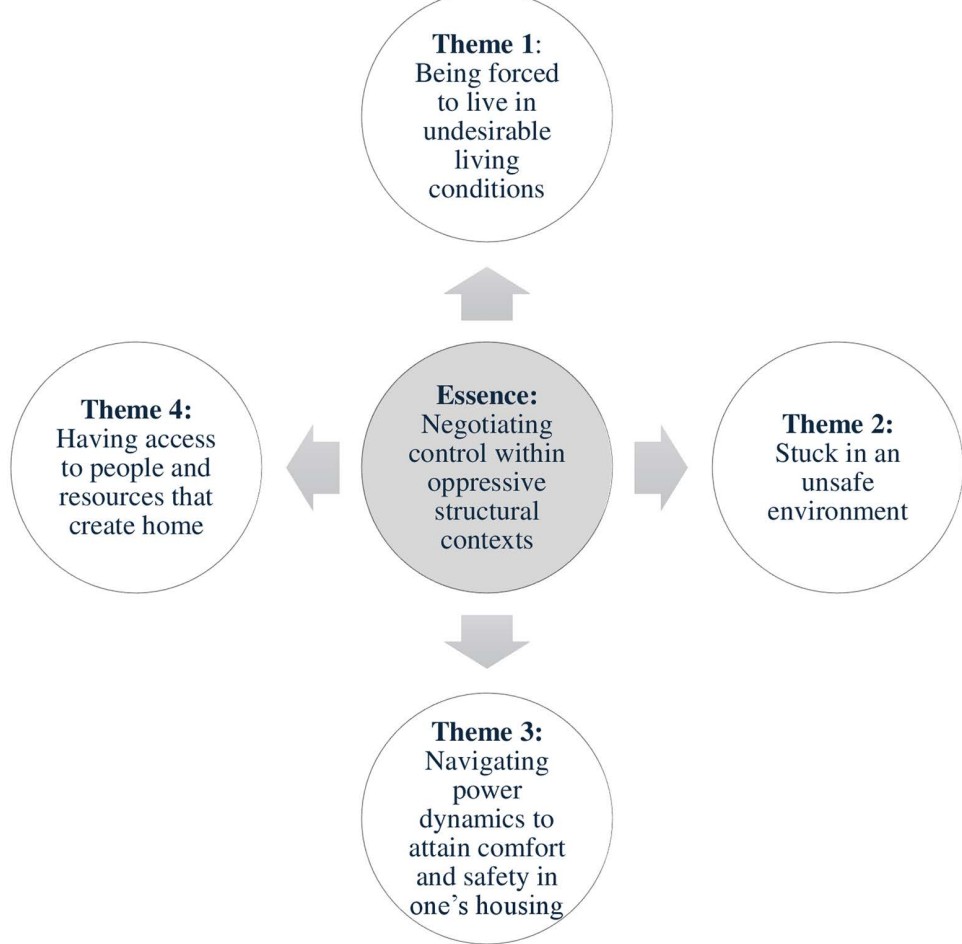

**Fig 1. Summary of Themes.**

Further to living with infestations, participants described how they were living in poorly maintained housing where there were issues with mould and broken furnishings that were not addressed by the landlord in a timely manner: "*things don't get fixed right away*" [Casey]. Further to living in poorly maintained units, some participants described living in over-crowded rooming houses and illegal suites. Although these were not considered to be ideal living situations, participants recognized that it was all that was available to them: "*that's what you get for $550... it's just like I want my own space, my own bachelors... my own kitchen*" [Alexander]. One participant described their experience of living in an illegal suite in housing that was overcrowded. The presence of a illicit substance trade in this overcrowded living situation contributed to these living situations feeling erratic, and interfered with their ability to live out their daily lives and engage in regular daily activities: "*I didn't want to do anything when I lived in a laundry room*" [Bruce]. Compounding this problematic situation, living in an illegal suite that was apparently sublet to him meant that Bruce's housing was always precarious. One day, another tenant in the rooming house in which he was living announced that he had to leave: "*I was told I could stay there and then... like a few days later the guy I was staying with told me that... the landlord was coming and that I had to move all my stuff out*" [Bruce].

**Theme 2: Stuck in an unsafe environment.** Several participants identified how they struggled to feel emotionally safe in their home and neighbourhood following homelessness. Michelle described how her "*neighbourhood isn't the best neighbourhood…I see people overdose and die out front of my window.*" Pekoe indicated that due to a limited income, he was forced to live in unsafe areas of the city in which he resided: "*I have to live in high-risk areas, with high-risk people because I am on an ODSP* [Ontario Disability Support Program] *budget.*" Safety was an ongoing concern for participants, who were exposed to interpersonal conflicts among other tenants in their housing on a regular basis, often due to living in rooming houses that were the only affordable housing option available within their budgets. Participants described that "*fighting among people, food theft, breaking into other people's room*[s]" [Doc] was the norm in rooming houses. These conditions caused participants to feel unsafe in rooming houses overall, and having no choice in who was also living in other rooms in the houses in which they were situated compounded these feelings: "*…a lot of the times the rooms were shared, which is normal for a group home, but I didn't really feel safe with a lot of people I was with*" [Gavin].

Participants described living in housing with an active illegal substance trade, which frightened them and exposed them to trauma: "*…the drugs, yep drugs. There was a fight. Some guy almost died. I got traumatized by that cause I heard it all happen. I seen it and it was frightening*" [Amber]. The housing available to individuals after leaving shelters and the street often involved exposure to other individuals who were actively engaged in problematic substance use, and participants highlighted their concern for individuals who were leaving homelessness and wanted to reduce or abstain: "*so say you have somebody who is homeless who is a severe drug addict…and you put them into housing and then you just drop them right away. They're just gonna relapse because of the community that you're putting them into*" [Michelle]. To individuals who were not using substances, their lives in shared accommodation with persons who were living with addiction caused their immediate environments to feel more chaotic than they wanted them to be. In describing a former roommate, one participant stated that: "*she used…coke so she sometimes would party for days on end and…just be constantly running around* [playing] *metal music and I don't know. It just wasn't, like there was a lot of disturbances in that house*" [Bruce]. Another participant remarked about his current roommate: "*his alcoholism is not my problem…but it is now that he's living with me because I have to tolerate this*" [Pekoe].

Beyond substance use, participants described situations where roommates would invite strangers into their housing. Having little knowledge of the backgrounds of their roommates' acquaintances caused them to feel exposed to some unwanted risks: "*she was just meeting men online to bring them home and it made me nervous. Like cause I'm sleeping in the room next* [door]. *Like what could happen, you know?*" [Casey]. The ongoing worry about whether they would be safe in these living conditions negatively influenced the mental health of participants following homelessness: "*I developed like a really bad paranoia*" [Gavin]; and "*it made me nervous*" [Casey].

**Theme 3: Navigating power dynamics to attain comfort and safety in one's housing.** Participants identified that the social contexts in which they were situated played an important role in determining the quality of their housing after homelessness. This included both the positive and negative control that roommates, neighbours, landlords, and staff could have over the quality of their housing. Participants described situations where their housing was always precarious and "*they could be kicked out at any point*" [Bruce].

Landlords and staff held particular power by having the ability to evict tenants and enforce rules in the housing in which participants were living. For some, the number of rules that they needed to follow in their housing led to feelings of oppression: "*transitional housing makes me feel like I'm in jail*" [Alexander]. Others described feelings of being under the constant surveillance and control of powerful others: "*…she* [landlord] *would immediately see me and then start heckling me or nagging me about something or threatening me*" [Gavin]. While for some, the control imposed by landlords and staff in permanent and transitional housing led to feelings of oppression, for others who were able to develop more positive relationships with their landlords, the feeling was more communal and supportive: "*it's been helping me cause it's a very relaxed place* [and] *the landlord is very, very nice*" [Perseus].

Due to a lack of affordable housing in their communities, participants were often given no choice but to live in undesirable buildings and neighbourhoods and with roommates with whom they could share the cost of housing. As such, they found themselves living around neighbours and with roommates that they wouldn't typically choose to live with, and conflicts emerged that often threatened their ability to sustain their tenancies. Roommates and neighbours were described as having the ability to enact power by influencing landlords and staff in their housing, described by one participant as "*pointing fingers*": "*Anything that happens in the house it's all hearsay, it's all pointing fingers*" [Alexander]. Power was exerted by roommates and neighbours by affecting participants' ability to feel comfortable in their housing: "*I feel like a guest in my own place*" [Gabriella]; and "*I wasn't really comfortable using the phone very often…my one roommate was always partying*" [Bruce].

Not only did roommates influence the psychological comfort and safety of participants' housing, but also influenced the experience of their housing through differences in how they cared for shared spaces. One participant described how they felt forced to eat outside of their housing because the kitchen was not kept clean: [my roommate] "*…makes a sandwich, and he just leaves the bread open on the table and the dirty knife… the kitchen is just awful*" [Doc]. While most participants described the negative influence of roommates, and conflicts that emerged as a result, some described positive experiences. Annabella reported, "*It's actually really good to have roommates, that's actually healthy for me.*" Other participants concurred, indicating that "*having good people around for support and stuff*" [Amber] was a protective factor for their mental well-being.

**Theme 4: Having access to people and resources that create home.** Participants described how having their own space, owning furniture and items to make their housing comfortable, being able to access the community, and having opportunities to be around other people all contributed to feeling that their housing was of good quality. A lack of financial means was an ongoing source of stress. Pekoe shared that after paying rent and utilities from his social assistance funding, there was little left for purchasing basic items such as toilet paper and food: "*the money that's left over doesn't even last two weeks.*"

Many participants described how they struggled to adapt to living in housing following homelessness, and that this process of adaptation was something that unfolded over time: "*I was still so used to…outside living and street life that it was hard for me to stay indoors…eventually, it became like home. I made it a home and I took the time and used the resources available…and I got furniture*" [Barbara]. Bruce described how the first housing that he had obtained following homelessness was an illegal suite, and that he had already moved once to be in a better living situation: "*I'm still getting on my feet…like income wise…I used to live in a laundry room…having my own space with my own door is a big difference…not having to worry about my stuff, being able to have stuff.*" Community agencies provided vouchers to purchase furniture and other household items that were helpful in making participants' housing feel like home, and facilitated social

connection: [when you have] "*proper furniture…you can have friends over whereas if you don't have furniture, you don't really wanna have your friends over*" [Michelle].

Participants who were living in rooming houses lacked privacy and space, which impaired their ability to feel settled. The desire to 'make home' by having pets and to be able to decorate their housing units was often prevented by their current living situations:

> *…not being able to…just decorate my room the way I wanted to. You know, not being able to…hang things on the walls…and not being able to…get a different desk that fits the space better…things like that have meant that I have felt like I am living in someone else's house. And it has not felt like home.* [Annabella]

When Annabella found a new place to live that was more suitable for her needs, the windows and the spaciousness of her bedroom brought hope by enabling her to engage in activities that improved her overall well-being, a sentiment expressed by other participants: "*I've really filled my place with things that make me happy…and make me comfortable. And even when I have bad days with my mental health, I'm still able to just be in this space.*" [Gavin].

Finally, participants described how they longed for opportunities to belong in their housing, and that this wasn't always happening in their current situation. Some participants were housed in a neighbourhood that was far away from their social networks resulting in feelings that their housing was "*very isolating and very depressing*" [Gavin]. Others had similar experiences: "*I kind of wanna be around people right now…It's then that I really struggle with being alone*" [Sunshine]. In contrast, some participants were in living situations where they were fortunate to be close to their social networks: *"…being where I am with people who care about me, I actually feel good*" [Michelle]. Michelle described how living with mental illness caused her to isolate, and how important being housed in an apartment that was close to her social networks was especially important for maintaining her well-being: "*it's really, really hard for me to isolate cause as soon as I isolate, I have someone banging on my door to come play cards.*" Casey described a similar experience in her apartment building: "*I've never had so much social life…since I moved here…when I come here it was…nice and I trust everybody here, which is a nice feeling.*"

## Discussion

We conducted this study to explore experiences of housing quality following homelessness for individuals living in two cities in Ontario, Canada. Participants included in this study had been housed for less than three years after leaving homelessness, and had varying experiences of housing quality. Many described being dissatisfied with their current situation because their housing quality was poor, but felt that they had few other options. Participants cited living on a limited income that prevented choice in what housing they could obtain, and many found themselves living in rooming houses and illegal suites to afford rent, a situation that they would not otherwise chooose. Living in these undesirable situations had a serious influence on participants' mental health for better or worse. When their housing was described as good quality, participants reported improved quality of life and mental well-being, a finding consistent with previous research [24]. For many, however, housing was described as poor in that participants felt unsafe and uncomfortable due to living in housing that was poorly maintained, with problematic roommates, or that was infested with vermin or bedbugs. This study builds on existing research exploring experiences of housing quality following homelessness [4] by providing evidence on the experience of housing quality across a range of housing contexts including permanent supportive housing, social housing and market housing.

This study is the first known to our team exploring the experience of housing quality following homelessness in a Canadian context [4], and provides an important glimpse into the housing conditions in which individuals who are leaving homelessness are situated after leaving shelters or the street. Our findings indicate that individuals leaving homelessness in a Canadian context frequently obtain housing that is sub-standard, and may lack the necessary conditions for attaining a

basic degree of well-being. If our goal as researchers, policymakers and practioners is to end homelessness and prevent ongoing homelessness [13], it is critical that we understand the quality of housing that is available to individuals following homelessness. Gaining an enhanced understanding of the conditions of a person's housing following homelessness, and how these conditions influence well-being across different countries and housing contexts can inform future policy and practice aimed at improving the state of housing for individuals living on the lowest incomes in society including persons leaving homelessness. This is an essential endeavour that will contribute to building a housing and support system aimed at truly preventing and ending homelessness. The housing available to participants in the current study was largely poor, and imposed a serious and negative influence on psychosocial well-being. While this study builds on existing literature by providing evidence from Canada and across a range of housing contexts, research on experiences of housing quality beyond Canada and the United States is needed.

The growth of homelessness across the globe over the past forty years is an international human rights travesty that will only be adequately addressed by broad scale, renewed government investment in social housing [36,37]. Concurrent with the development of this international problem, housing markets in most countries have continued to rise unabated, while at the same time, governments across the globe have significantly decreased investment in social housing [38]. These developments have both caused and deepened this housing and homelessness crisis. During this time, instead of building new social housing to meet the needs of citizens living in low income, governments have instead begun to rely on the market system to provide housing to individuals living on the lowest incomes in society. The significant rise in homelessness across the globe proves that this approach has been nothing short of disastrous. Furthermore, the significant rise in homelessness, coupled with a lack of housing overall has created the conditions wherein the housing available to individuals living in low income is often of poor quality, which has placed persons who are leaving homelessness at risk. Policymakers should be aware of the impact of existing housing policies on the development and maintenance of homelessness across the globe, and how the existing reliance on the market system has impacted on the lives of persons who are leaving homelessness. To prevent and end homelessness, it is imperative that policymakers devise policies that mitigate the financialization of housing, and result in the restoration of the social housing system in Canada and beyond [38,39]. Further, policies that protect tenants from living in poorly maintained housing need to be revised, and more consistently enforced to ensure that individuals who are leaving homelessness are able to access good quality, affordable housing in which they can thrive.

Practitioners who are supporting persons who experience homelessness should be aware of the housing situations in which service users are living, and account for thes as they provide health and social care. While practitioners who work in community-based settings are often aware of the conditions in which service users live and experience high degrees of moral distress as a result [2], those working in outpatient or inpatient roles may have a limited understanding of the quality of housing in which the individuals they support are situated. Our findings reveal the influence of housing quality on the mental well-being of persons who are leaving homelessness. The stress and subsequent mental health challenges caused by living in poor quality housing may be the reason for accessing services, and practitioners may consider discussing the quality of one's housing when conducting their initial and ongoing assessments over the course of care. Goals related to accessing good quality housing should be prioritized given the ways in which it can influence health and well-being.

## Limitations

This study was conducted with participants leaving homelessness in a high-income country, and our findings should be interpreted with that context in mind. Experiences of housing quality following homelessness in low or middle-income countries are likely to differ significantly from the participants in the current study. Further, while we tried to recruit participants of a range of ages, races, genders and sexual orientations, we were unable to recruit a sample as diverse as we wished given a lack demographic representation of key groups in the cities in which we recruited. As such, the participants

in the current study lacked diversity with respect to race and sexual orientation, and any interpretation of our findings should account for the fact our sample was primarily white and heterosexual. Individuals conducting future studies should seek participation of these underrepresented groups in communities with greater diversity with respect to race and sexual orientation. While we would have preferred to engage in member-checking with the participants following analysis, we could not do so in the current study, thereby limiting our ability to confirm with participants whether our analysis truly reflected their experiences as they understood them. Further, it is well recognized that secondary analyses are limited in that the research question used for analysis differs from the original research question [40]. Readers should be aware that the findings of the current study are limited by the fact that our research question did not focus on housing quality specifically, and important experiences of housing quality following homelessness may not have been identified in our findings. Future research focused specifically on experiences of housing quality following homelessness may provide a more fulsome understanding of this phenomenon to build on the analysis presented in the current study.

## Conclusion

Preventing and ending homelessness is a critical objective that can protect the health and well-being of individuals who live in poverty across the globe. Housing quality is an important determinant of well-being, and attaining housing that meets a basic living standard is essential for supporting tenancy sustainment and thriving following homelessness. In a country as wealthy as Canada, many would assume that the quality of housing available on the rental market would be of good quality. Our findings indicate that for individuals living on the lowest incomes in society, housing of good quality is often elusive. This situation represents a serious public health and human rights problem that requires collaboration among persons with lived experience, researchers, policymakers and practitioners to solve. In so doing, there is potential for making important contributions towards building a resilient system that is structured to reach the important goal of preventing and ending homelessness.

## Acknowledgments

We would like to thank the participants in this research who shared important perspectives that we hope will influence research, policy and practice in services designed to support persons who have experienced homelessness. We are touched by your trust and the candour demonstrated in sharing your important experiences. We also want to acknowledge the valuable feedback received from peer reviewers and the editorial team during the publication process. We recognize and appreciate the time and energy dedicated to this process.

## Author contributions

**Conceptualization:** Carrie Marshall, Jessica Allen, Corinna Easton, Rebecca Goldszmidt, Elham Javadizadeh, Shauna Perez, Brooklyn Ward.

**Data curation:** Carrie Marshall.

**Formal analysis:** Carrie Marshall, Patti Plett, Jessica Allen, Corinna Easton, Rebecca Goldszmidt, Elham Javadizadeh, Shauna Perez, Brooklyn Ward.

**Funding acquisition:** Carrie Marshall.

**Investigation:** Carrie Marshall.

**Methodology:** Carrie Marshall.

**Project administration:** Carrie Marshall.

**Resources:** Carrie Marshall.

**Software:** Carrie Marshall.

**Supervision:** Carrie Marshall.

**Validation:** Carrie Marshall.

**Visualization:** Carrie Marshall.

**Writing – original draft:** Carrie Marshall, Patti Plett, Jessica Allen, Corinna Easton, Rebecca Goldszmidt, Elham Javadizadeh, Shauna Perez, Brooklyn Ward.

**Writing – review & editing:** Carrie Marshall, Patti Plett, Jessica Allen, Corinna Easton, Rebecca Goldszmidt, Elham Javadizadeh, Shauna Perez, Brooklyn Ward.

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
