## [Decision Letter · Decision Letter 0]

17 Dec 2024

PMEN-D-24-00507

“I just don’t know what will be better other than an apartment which I can’t afford”: Experiences of Housing Quality Following Homelessness in Ontario, Canada

PLOS Mental Health

Dear Dr. Marshall,

Thank you for submitting your manuscript to PLOS Mental Health. After careful consideration, we feel that it has merit but does not fully meet PLOS Mental Health’s publication criteria as it currently stands. Therefore, we invite you to submit a revised version of the manuscript that addresses the points raised during the review process.

EDITOR: Thank you for submitting your manuscript to PLOS Mental Health. Please read the very positive reviews carefully, revise your manuscript according to their suggestions to further improve the quality, and upload the revised version.

We look forward to receiving your revised manuscript.

Kind regards,

Paolo Raile

Academic Editor

PLOS Mental Health

Journal Requirements:

2. Please provide separate figure files in .tif or .eps format.

https://journals.plos.org/mentalhealth/s/figures 

https://journals.plos.org/mentalhealth/s/figures#loc-file-requirements 

Additional Editor Comments (if provided):

Reviewers' comments:

Reviewer's Responses to Questions

**Comments to the Author**

1. Does this manuscript meet PLOS Mental Health’s publication criteria ? Is the manuscript technically sound, and do the data support the conclusions? The manuscript must describe methodologically and ethically rigorous research with conclusions that are appropriately drawn based on the data presented.

Reviewer #1: Yes

Reviewer #2: Yes

2. Has the statistical analysis been performed appropriately and rigorously?

Reviewer #1: N/A

Reviewer #2: N/A

3. Have the authors made all data underlying the findings in their manuscript fully available (please refer to the Data Availability Statement at the start of the manuscript PDF file)?

Reviewer #1: No

Reviewer #2: No

4. Is the manuscript presented in an intelligible fashion and written in standard English?

Reviewer #1: Yes

Reviewer #2: Yes

5. Review Comments to the Author

Reviewer #1: Thank you for asking me to review this well written paper that addresses an important human rights problem in our society. I appreciate the prolonged engagement of the researchers with the population of interest, which can be a hard to reach group. The conclusions are sad but important to expose.

I have a few minor suggestions:

The section on recruitment of participants is somewhat unclear. Ethics approval was obtained by the authors of the parent study, but also for the present one?

Only a subsample of participants from the initial sample was included in the secondary analysis presented in the current paper. Do I understand correctly they were asked for consent for the secondary analyses of the transcripts of the original study? Please clarify this.

The described analysis matches inductive coding, what do the researchers specifically mean by abductive coding?

The strategies of trustworthiness mostly pertain to the parent study, the authors should mention this.

A few sentences about ethics in the method section might be helpful (in addition to the attachment). For example, did the participants receive anything for their participation and how did the researchers do justice to their efforts? Are results communicated back to participants?

In the results section a table picturing the themes at a glance would be helpful.

In the results section, quotes come with names of participants. Are these their real names? Please clarify (in the text) the choices made in this.

Did researchers in the parent study do a member check to increase validity? If not, this could be mentioned as a limitation. The fact that data were not collected on the basis of the present research question and that analyses were secondary, should also be mentioned. What are possible implications of this?

Reviewer #2: I would like to begin by congratulating the team on this paper. It addresses an important and relevant problem in your context and highlights the need for more research led by individuals with lived experiences in this area. I believe this work is purposeful and has significant potential for impact. However, there are a few areas that could benefit from further clarification, as outlined below:

Methods Section

1. Interviewer Characteristics

It would be helpful to provide more information regarding the interviewers’ qualifications. Were all interviewers properly trained, or did they have any prior experience conducting interviews of this nature?

2. Participant Selection

-While the sample size appears optimal, more explicit detail on the sampling method is needed. What type of sampling strategy was employed, and what was the rationale behind it? - There is a noticeable lack of diversity in the sample. The participants are predominantly white, with only two African-American individuals included. Considering that social experiences between racial groups may differ significantly, addressing this limitation or discussing its potential impact on the findings would be beneficial.

-Although I understand that full recruitment details are available in the parent study, i suggest it would be useful to specify how participants were approached for these interviews. Additionally, were there any participants who were approached but declined or were excluded?

3. Interview Setting

Additional details about the interview process would provide greater clarity. Specifically:

-Where and how were the interviews conducted?

-Were all interviews conducted in person? If not, what were the alternative methods (e.g., virtual or phone interviews)?

-On average, how long did the interviews last?

-Could you describe the structure of the interviews and provide more detail about the questionnaire, was a guide used?

-It is noted that participants’ names or pseudonyms were used to identify verbatim quotes. While this approach is not common, it would be useful to explain the rationale behind this choice and how it was managed to maintain anonymity and confidentiality.

Results: Its unclear which are major and minor themes

6. PLOS authors have the option to publish the peer review history of their article (what does this mean? ). If published, this will include your full peer review and any attached files.

**Do you want your identity to be public for this peer review?** For information about this choice, including consent withdrawal, please see our Privacy Policy .

Reviewer #1: No

Reviewer #2: No

---

## [Decision Letter · Decision Letter 1]

26 Feb 2025

PMEN-D-24-00507R1

“I just don’t know what will be better other than an apartment which I can’t afford”: Experiences of Housing Quality Following Homelessness in Ontario, Canada

PLOS Mental Health

Dear Dr. Marshall,

Thank you for submitting your manuscript to PLOS Mental Health. After careful consideration, we feel that it has merit but does not fully meet PLOS Mental Health’s publication criteria as it currently stands. Therefore, we invite you to submit a revised version of the manuscript that addresses the points raised during the review process.

EDITOR: Thank you again for your submission. Both reviewer are nearly fully satisfied. Both reviewer mentioned just little minor changes after which the article can be accepted. Please revise the manuscript to meet reviewer's comments. Thank you!

We look forward to receiving your revised manuscript.

Kind regards,

Paolo Raile

Academic Editor

PLOS Mental Health

Journal Requirements:

Additional Editor Comments (if provided):

Reviewers' comments:

Reviewer's Responses to Questions

**Comments to the Author**

1. If the authors have adequately addressed your comments raised in a previous round of review and you feel that this manuscript is now acceptable for publication, you may indicate that here to bypass the “Comments to the Author” section, enter your conflict of interest statement in the “Confidential to Editor” section, and submit your "Accept" recommendation.

Reviewer #1: All comments have been addressed

Reviewer #2: All comments have been addressed

2. Does this manuscript meet PLOS Mental Health’s publication criteria ? Is the manuscript technically sound, and do the data support the conclusions? The manuscript must describe methodologically and ethically rigorous research with conclusions that are appropriately drawn based on the data presented.

Reviewer #1: Yes

Reviewer #2: Yes

3. Has the statistical analysis been performed appropriately and rigorously?

Reviewer #1: N/A

Reviewer #2: Yes

4. Have the authors made all data underlying the findings in their manuscript fully available (please refer to the Data Availability Statement at the start of the manuscript PDF file)?

Reviewer #1: No

Reviewer #2: Yes

5. Is the manuscript presented in an intelligible fashion and written in standard English?

Reviewer #1: Yes

Reviewer #2: Yes

6. Review Comments to the Author

Reviewer #1: Dear authors,

Thank you for your careful consideration of my comments.

I have one more minor suggestion, it's about this statement:

“A separate ethics application was not required for the current study, as participants in the parent study provided approval for the conduct of secondary analyses of anonymized transcripts. In the parent study, participants were recruited from organisations providing social service and mental health support to persons with lived experiences of homelessness …”

Since anonymity cannot be guaranteed (someone can always be recognized by the story), I would speak of ‘pseudonymization’.

Congratulations on this important paper.

Reviewer #2: Thank you for your revisions. Authors have addressed all my observations thoroughly and the manuscript is now stronger.

I suggest some minor revisions be considered in methods section:

-Please consider including that AUDIT and DAST scales were part of the data collection tools.

-Analysis: Please revise if all citations are in vancouver format.

7. PLOS authors have the option to publish the peer review history of their article (what does this mean? ). If published, this will include your full peer review and any attached files.

**Do you want your identity to be public for this peer review?** For information about this choice, including consent withdrawal, please see our Privacy Policy .

Reviewer #1: No

Reviewer #2: No

---

## [Editor Report · Decision Letter 2]

14 Mar 2025

“I just don’t know what will be better other than an apartment which I can’t afford”: Experiences of Housing Quality Following Homelessness in Ontario, Canada

PMEN-D-24-00507R2

Dear Dr. Marshall,

We are pleased to inform you that your manuscript '“I just don’t know what will be better other than an apartment which I can’t afford”: Experiences of Housing Quality Following Homelessness in Ontario, Canada' has been provisionally accepted for publication in PLOS Mental Health.

Best regards,

Paolo Raile

Academic Editor

PLOS Mental Health